

# A super-resolution network using channel attention retention for pathology images

Feiyang Jia[1], Li Tan[1], Ge Wang[1], Caiyan Jia[2] and Zhineng Chen[3]

[1] Beijing Key Laboratory of Big Data Technology for Food Safety, Beijing Technology and Business University, Beijing, China
[2] School of Computer and Information Technology, Beijing Jiaotong University, Beijing, China
[3] School of Computer Science, Fudan University, Shanghai, China

## ABSTRACT

Image super-resolution (SR) significantly improves the quality of low-resolution images, and is widely used for image reconstruction in various fields. Although the existing SR methods have achieved distinguished results in objective metrics, most methods focus on real-world images and employ large and complex network structures, which are inefficient for medical diagnosis scenarios. To address the aforementioned issues, the distinction between pathology images and real-world images was investigated, and an SR Network with a wider and deeper attention module called Channel Attention Retention is proposed to obtain SR images with enhanced high-frequency features. This network captures contextual information within and across blocks via residual skips and balances the performance and efficiency by controlling the number of blocks. Meanwhile, a new linear loss was introduced to optimize the network. To evaluate the work and compare multiple SR works, a benchmark dataset bcSR was created, which forces a model training on wider and more critical regions. The results show that the proposed model outperforms state-of-the-art methods in both performance and efficiency, and the newly created dataset significantly improves the reconstruction quality of all compared models. Moreover, image classification experiments demonstrate that the suggested network improves the performance of downstream tasks in medical diagnosis scenarios. The proposed network and dataset provide effective priors for the SR task of pathology images, which significantly improves the diagnosis of relevant medical staff. The source code and the dataset are available on https://github.com/MoyangSensei/CARN-Pytorch.

## INTRODUCTION

The single image super-resolution (SISR) (*Wang, Chen & Hoi, 2020*) is a task that uses low-level features to reconstruct a high-resolution (HR) image from a low-resolution (LR) one. It has been widely used in surveillance equipment, satellite remote sensing, medical imaging, and other fields (*Farooq et al., 2020*; *Li et al., 2020*; *Chen et al., 2020*; *Zhang et al., 2018a*; *Falahkheirkhah et al., 2019*). Since each input image usually corresponds to multiple

Corresponding author
Li Tan, tanli@th.btbu.edu.cn

reasonable output images in the SISR tasks, the tasks are defined as a typical ill-posed inverse problem (*Yoo, Lee & Kwak, 2018*). In this context, traditional SISR methods (*Kim, Bose & Valenzuela, 1990*; *El-Khamy et al., 2005*; *Li & Orchard, 2001*; *Irani & Peleg, 1991*; *Hardie, Barnard & Armstrong, 1997*) have focused on using a large amount of image prior information, and have made key achievements. In the past 20 years, as convolutional neural networks (CNNs) have been developed rapidly and achieved great success in computer vision, a large number of CNN-based models have also emerged in SISR. These models generate LR images by applying a degradation model (such as bicubic down-sampling) according to existing HR images to form HR-LR pairs and subsequently concentrate on developing and training models to generate high-quality HR images.

The majority of deep learning-based SISR models were initially designed for real-world images. As a pioneer work, *Dong, Loy & Tang (2016)* proposed a SISR model SRCNN, that only used three convolutional layers to build a sparse coding-based convolutional network, ushering in the SISR deep learning era. *Ledig et al. (2017)* proposed SRGAN, which was the first use of Generative Adversarial Networks (GAN) to address the SISR problem, emphasizing the importance of subjective metrics in the evaluation of a SISR model. After the self-residual structure (*He et al., 2016*) was suggested and widely used in various deep learning models, *Lim et al. (2017)* proposed EDSR/MDSR models, which extended the residual structure in SRResNet (*Ledig et al., 2017*) for SISR tasks. Afterwards, *Zhang et al. (2018b)* suggested the model RCAN , which adopted an attention mechanism to select different feature channels. Based on the residual network, *Zhang et al. (2018c)* introduced a dense network RDN to further improve the performance of SISR on real-world images.

Furthermore, with the widespread interest in medical digital diagnostics, SISR models have been developed and used in processing medical images. *Srivastav, Gangi & Padoy (2019)* integrated 2D human pose estimation into the super-resolution (SR) Network to address the privacy concerns of cameras installed in operating rooms. *Zhao et al. (2019)* proposed a model named CSN for HR Magnetic Resonance Imaging (MRI) images. *Qiao et al. (2021)* constructed a dataset BioSR and trained a network named DFCAN/DFGAN for scientific microscope images. They used the SSIM loss as a part of the optimization strategy and pointed out the role of SSIM in medical image reconstruction. *Li et al. (2019)* suggested a two-stage SR network for HR reconstruction of arterial spin-labeled perfusion MRI images. *Chen et al. (2020)* proposed SWD-Net to SR histopathology images using the wavelet transform. They used two interpolation algorithms as degenerate kernels: bicubic and nearest neighbor. Using bicubic enables SWD-Net to exhibit better performance. *Mukherjee et al. (2019)* reconstructed HR images from intermediate-resolution images at multiple levels in whole slide imaging (WSI). Based on SRGAN, *Shahidi (2021)* suggested the Wide Attention SRGAN (WA-SRGAN) and applied improved Wasserstein with gradient penalty to stabilize the model during training.

Although existing SR methods have achieved significant results on objective metrics such as Peak Signal to Noise Ratio (PSNR) (*Hore & Ziou, 2010*) and Structural Similarity (SSIM) (*Hore & Ziou, 2010*; *Wang et al., 2004*), they relied on large and complex network structures to improve network performance in terms of their depth or width. Taking

RCAN (*Zhang et al., 2018b*) as an example, its baseline network uses 10 residual modules, each of which contains 20 residual blocks based on channel attention, and even the overall convolutional layer depth exceeds 1,000 layers. In medical diagnosis scenarios, large-scale SR Networks inevitably make the prediction results extremely smooth and blur the edges of a generated HR image for reaching distinguished objective metrics (*Wang, Chen & Hoi, 2020*), resulting in the loss of major image details and disturbing the decision-making of diagnosticians or diagnostic algorithms. Simultaneously, a deeper network generates more computation costs, which is not allowed in medical diagnosis scenarios with extremely tight timelines. Multiple SISR models in other fields failed to achieve a satisfactory balance between efficiency and performance when applied to medical diagnosis due to this contradiction. Furthermore, since the high and low-frequency information distribution of pathology images differs greatly from real-world images, SR Networks or pre-trained models—which are not designed for pathology images—are not able to achieve the best performance results for pathology images in SR tasks, obviously. Besides, the majority of the existing SR Networks were built on a single backbone structure with no skips (*Wang, Chen & Hoi, 2020*), making it difficult to capture multi-scale contextual information.

Recently, residual structures have been used in an increasing number of image reconstruction tasks. A large body of works has demonstrated the superior performance of residual structures in SISR tasks (*Ledig et al., 2017*; *Lim et al., 2017*; *Zhang et al., 2018b*; *Zhang et al., 2018c*; *Kim, Lee & Lee, 2016*; *Dai et al., 2019*). Compared with other structures, residual connections in a network are easier to learn and optimize, making the optimization of an extremely deep SR Network possible. Furthermore, the Channel Attention (CA) mechanism enables a network to identify target areas while scanning a global image and subsequently invest additional computing resources in these target areas to suppress redundant information. This mechanism has been adopted and proved to be useful in multiple SISR models (*Zhang et al., 2018b*; *Mei et al., 2020*; *Dai et al., 2019*).

To balance the efficiency and accuracy of a SISR model for medical pathology images, this study proposes a deep residual structure-based module called the Channel Attention Retention Block (CARB), which extracts as much feature information as possible in a limited block depth through multiple convolutions with wider and deeper attention channels. Afterwards, a small number of CARBs is used to form the Channel Attention Retention Network (CARN), in which inter-block residuals, as well as intra-block residuals, are used to compensate for the performance reduction caused by the decreasing network depth. To fully utilize the ability of the attention mechanism in the CARN network, a combined loss function with MSE loss, L1 loss, and SSIM loss is specifically designed (*Wang et al., 2004*). To force this network to pay further attention to wider and more critical regions in a pathology image, a new dataset named bcSR is created by collecting and filtering breast cancer tissue lymphatic slice images from the public dataset CAMELYON (*Litjens et al., 2018*).

The contributions of this study are summarized as follows: (1) A benchmark dataset bcSR is constructed; it contains 1,200 images with a large pixel-mean difference of RGB channels. The dataset makes SR models focus on a wider range of regions. Moreover, it does not only provide reliable data for SR tasks of breast cancer pathology images but is

also used as input for general regression tasks, such as cancer region detection. (2) An SR Network CARN is developed based on residual structures and the attention mechanism to balance its efficiency and accuracy on the SISR tasks for pathology images. According to the nature of pathology images and the task orientation of the network, a combined loss function is designed based on the MSE loss, L1 loss, and SSIM loss to improve the overall performance of the network. The superior performance and efficiency of CARN are demonstrated by multiple contrast experiments.

## RELATED WORK

As mentioned in the previous section, *Ledig et al. (2017)* used residual structures in SRGAN for the first time and built the SRResNet as their network's generator. The SRResNet enables the network to support the inpainting of higher frequency image details, compensating the decrease in objective metrics caused by SRGAN's perceptual loss. They pointed out that residual connections effectively ensure gradient information propagation in deep networks and enhance the robustness of GANs. *Kim, Lee & Lee (2016)* introduced residual networks to train deeper SR Network structures and achieved an outstanding performance. They specifically mentioned that skip connections could reduce the burden of carrying identity information in an SR Network. In this content, nested skip connections significantly reduce the computational load of the corresponding network, allowing its quick convergence. Inspired by the SRGAN, *Lim et al. (2017)* constructed EDSR by removing some unnecessary modules in the residual structure, such as BN layers. The residual structure is usually used on high-level vision tasks. Thus, they proposed that when the residual structure is used on low-level tasks, such as SR, deep learning networks do not need a big number of network layers to achieve the expected results. However, the EDSR still has a large number of parameters compared to the contemporaneous research work.

In fact, directly stacking residual modules similar to EDSR has limited contribution for improving objective metrics. This is due to the fact that stacking modules do not utilize the features in a deeper network in a better way, even though the increase in network depth may improve the network's data mining capabilities. *Zhang et al. (2018b)* adopted the Residual in Residual (RIR) structure in RCAN, so that the network would not lose a lot of its ability to represent features as it grows deeper. *Dai et al. (2019)* suggested a Second-Order Attention Network (SAN) with a Share-Source Residual Group (SSRG) core structure to capture long-range spatial context information. The SSRG added the LR features of the top layer in the network to the input of each basic residual group. This connection could effectively transfer the rich low-frequency information in LR images to the network's depth.

It is known that the CA mechanism has been widely used in deep neural networks in recent years, greatly improving the performance of corresponding networks in various tasks. *Hu, Shen & Sun (2018)* claimed that the output of a convolutional layer does not consider the dependence of each channel, and the CA allows a network to selectively enhance feature channels with large amounts of information. *Zhang et al. (2018b)* firstly employed the CA mechanism in SISR tasks. They assumed that the feature maps of different

channels in the network contributed differently to the reconstruction of high-frequency information in an image, and the difference between channels was necessarily increased so that the network could make full use of the enhanced channels while suppressing other ones. *Dai et al. (2019)* added a Second-Order Channel Attention module (SOCA) to the end of their base module besides a regular CA module in the whole model SAN, which used covariance normalization to explore the attention of second-order feature statistics. They believed that this module was able to explore the intrinsic correlation of features in a more effective way, thereby improving the expressive ability of CNNs.

Furthermore, this study focuses on the role of the CA mechanism for SR reconstruction of pathology images. The existing CA-based SR work ignores the information loss caused by CA, while acknowledging the significant contribution of CA to feature map reconstruction. This is because these models have a limited receptive field. Specifically, when there is a small amount of input information, the weight assigned by the attention mechanism is closer to the average value, causing features to be lost or fewer features to be caught. Hence, drawing diagnostic conclusions based on the status of a single cell in a medical diagnosis scenario is typically difficult because multiple regions of an image are usually correlated and the similarity of adjacent cell tissues is extremely high. One solution is non-local attention (*Wang et al., 2018*), which can adequately achieve the long-distance dependence between pixels. *Mei et al. (2020)* argued that cross-scale block similarity exists widely in natural images and proposed the Cross-Scale Non-Local (CS-NL) Attention module to compute pixel-to-block and block-to-block similarity within an image. However, this makes the calculation amount - of the network - increase rapidly. This study intends to design a new model to significantly reduce the information loss induced by the CA while retaining the CA's performance improvement.

## METHOD

### Overall structure

The overall architecture of the proposed model CARN is shown in Fig. 1. It was mainly constructed using a limited number of Channel Attention Retention Blocks (CARBs).

First, for each HR image $I_{HR}$, the corresponding LR image was generated $I_{LR}$ using the bicubic algorithm. Most SISR works choose Bicubic as the main degeneration kernel (*Lim et al., 2017*; *Zhang et al., 2018b*; *Zhang et al., 2018c*), and Bicubic is closer to the true degeneration of histopathological digital images (*Chen et al., 2020*). Subsequently, these images were segmented into fixed-size patches as the inputs of CARN. Considering that pathology images from a whole slide image (WSI) have various image styles, such as converted RGB images differentiated from the color structure caused by the staining of physical sections and scanning machines, the network has begun by normalizing the images by channels. Specifically, for an image $I_n$ with size of $H \times W$ in the dataset $I$ containing $N$ images, the normalization result $I_n^{norm}$ is obtained in Eq. (1):

$$I_n^{norm} = \sum_{i=1}^{H}\sum_{j=1}^{W}\left[ I_{n(i,j)}^{R} - I_N^{Rmean}, I_{n(i,j)}^{G} - I_N^{Gmean}, I_{n(i,j)}^{B} - I_N^{Bmean} \right] \qquad (1)$$

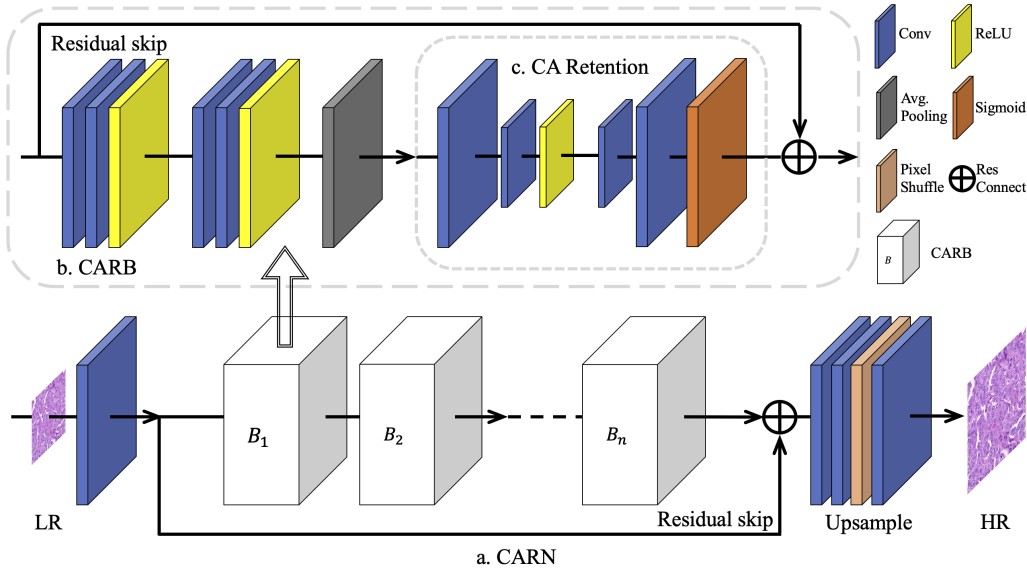

**Figure 1** **The overall architecture of CARN.** To overcome the challenge of training different multiples of SR networks on breast cancer pathology images, a basic network module is proposed based on residual connections, namely the Channel Attention Retention Block (CARB). Its end contains an attention submodule with wider internal channels and the capability of lingering attentional information, called CA Retention, which is able to handle inputs with arbitrary aspect ratios and resolutions. A fixed number of CARBs is used for concatenation to construct the CARN. Progressively upscaled convolutional layers and sub-pixel shuffling layers are employed to generate results for multiple up-sampling tasks.

where $\left[I_{n(i,j)}^{R}, I_{n(i,j)}^{G}, I_{n(i,j)}^{B}\right]$ are the R, G, and B value of image $I_n$ at position (i, j), and $\left[I_N^{Rmean}, I_N^{Gmean}, I_N^{Bmean}\right]$ is the mean value of RGB channel of dataset $I$.

Subsequently, the data went through several CARBs, where a long-skip connection with a residual structure was included, inspired by *Zhang et al. (2018b)*. It is found that simply stacking CARBs does not enable the network to perform better. The purpose of the long jump is to send the unconvolved low-level features in the image to the bottom of the network, supplementing the low-frequency information ignored by the multi-layer residual inside a CARB. At the end of the network, to generate $I_{HR}$ with the required scale, the sub-pixel shuffle layer (*Shi et al., 2016*) is used to amplify the features. For the SR tasks of 2×, 4×, and 8×, we specify the magnification of the sub-pixel shuffle layer as 2 and build the upsampling module using 1, 2, or 3 groups of such layers, respectively. For the 3× SR task, the number of features in multiple channels firstly increases to 9 times, and then the features pass through 1 group of sub-pixel shuffle layers with magnification of 3.

## Channel Attention Retention Block (CARB)

Previous CNN-based SR methods treated all feature channels as equal, which is not in line with reality. Each filter in a convolutional layer has a local receptive field of the same size, and its output does not contain any contextual information, as shown in Fig. 2A. Thus, *Zhang et al. (2018b)* used the CA mechanism to resolve this contradiction, as shown in Fig. 2B. The existing CA methods usually assign different weights to each feature channel,

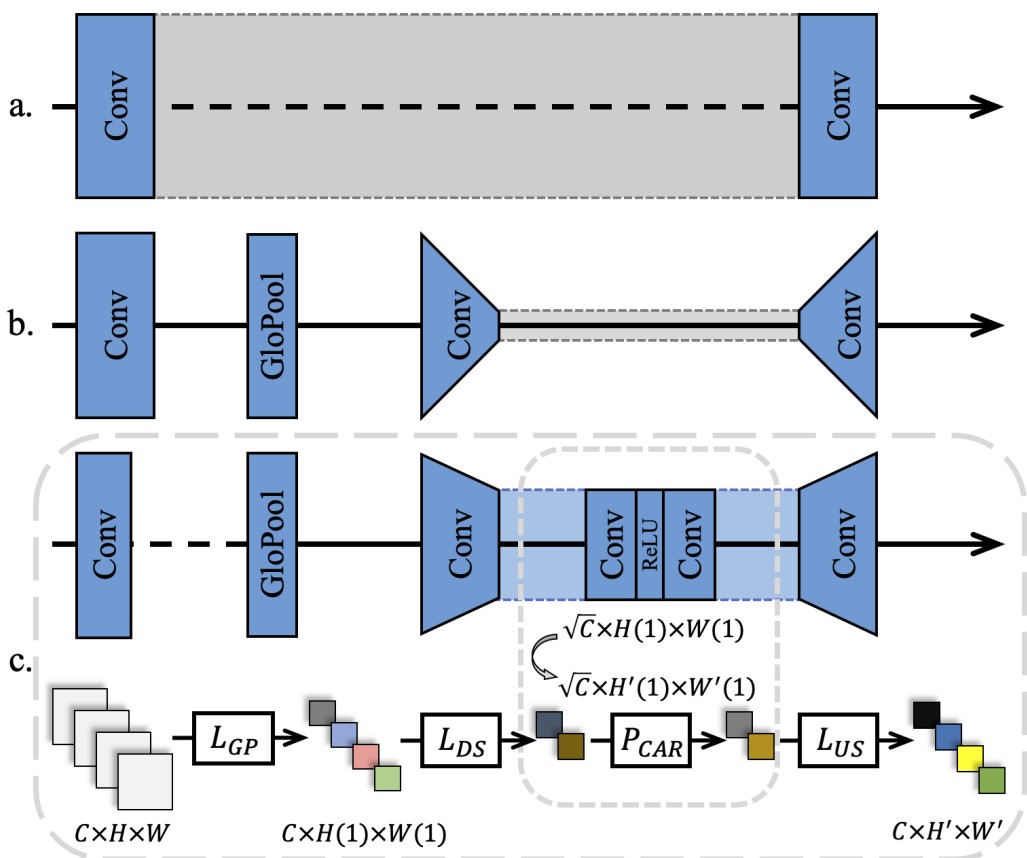

**Figure 2** **Different channel attention mechanisms.** (A) Networks without attention usually use a single feature width throughout the training process. (B) A narrow down-sampling convolutional layer was used to extract channel features. (C) A wider down-sampling convolutional layer for the CA retention was employed with the same channel width to extract further features after down-sampling.

while focusing on how to build a CA block using the spatial distribution of breast cancer pathology images. It is observed that high-frequency information in $I_{LR}$ derive from the palisade or honeycomb arrangement of human organs for pathology images. Therefore, pathology images contain richer high-frequency information and more evenly distributed low-frequency information than real-world images. Pathology images in RGB format converted from a digital pathology images format suitable for medical imaging devices, have a narrower range of color distribution than real-world images due to the staining operation of cell sections.

As a result, in this study, a global average pooling was adopted to convert the channel-related global spatial information into channel descriptors, and use Sigmod as the end-gating function. Meanwhile, two convolutional layers were added to further strengthen the up-sampling and the down-sampling features after the features were pooled as $1 \times 1$. Subsequently, a ReLU (*Nair & Hinton, 2010*) layer was added between the two convolutional layers to further activate the gating result, as shown in Fig. 2C.

To preserve the beneficial features of a pathology image as much as possible in a CARB and reduce the loss caused by channel compression, further down-sampling channels were set in the proposed model. In detail, the number of channels $C$ became $\sqrt{C}$ after the features were down-sampled. From the observation of most of the computer vision tasks using the CA mechanism, it is found that the value of *Reduction* significantly affects the training speed of deep networks. Although a minor *Reduction* value reduces the information loss caused by channel compression, it is not conducive to the convergence of a network and the stimulation of beneficial channels in the original feature map. On the other hand, increasing the *Reduction* value diminishes the computational load of a network to a certain extent. Hence, a wider down-sampling convolutional layer than the traditional CA mechanism was chosen to perform weight excitation and learn the weights in a wider feature distribution. This is due to the fact that SR tasks for pathology images require to average the weights over channels to offset the effects of the low inter-pixel variance and low-frequency regions that account for the vast majority of the areas in an image on features. In the following section, multiple ablation experiments demonstrate the rationality of the *Reduction* value.

Let the feature map $F = (f_1, \ldots, f_x, \ldots, f_C)$ be the input features, that means the feature map $H \times W$ with depth $C$. The average pooling layer shrinks $F$ to $1 \times 1$ to get the main representation of this channel $f_g$, as defined in Eq. (2):

$$f_g = \text{AvgPool}(f_x) = \frac{1}{H \times W} \sum_{i=1}^{H} \sum_{j=1}^{W} f_x(i,j) \tag{2}$$

where $f_x(i,j)$ represents the value of the $(i,j)$ position in the x-th two-dimensional feature $f_x$, and *AvgPool*(.) denotes the global average pooling. In a CA Retention, feature up-sampling and feature down-sampling are respectively performed on the main representation of all channels $F_g$, to obtain the attention-retained feature $F_{CAR}$, as defined in Eq. (3):

$$F_{CAR} = \text{Sig}\left(\text{FeatUS}\left(\delta\left(\text{FeatDS}(F_g)\right)\right)\right) \tag{3}$$

where $\delta(.)$ and *Sig*(.) represent the ReLU activation and Sigmod activation, respectively. *FeatDS*(.) and *FeatUS*(.) denote down-sampling convolution and up-sampling convolution for CA-retained features, respectively. The sampling multiples are set to $\sqrt{C}$ to get more learning caps for the CA Retention. This is different from existing SISR models for real-world images. Using the same settings as this study will not lead to a significant improvement in SR tasks for real-world images, but will introduce more information loss. At the end of the block, the residual component $F_{CAR}$ will be adaptively re-amplified.

In each CARB, the residual connection is introduced from the beginning to the end of the block, so that each CARB inherits and fully utilizes the residual information brought by the previous basic module. Let the initial feature vector of the network be $F_0$, and the transfer process of residual information is defined in Eq. (4):

$$F_m = R_m(F_{m-1}) = R_m(R_{m-1}(F_{m-2})) = R_m(R_{m-1}(\ldots R_1(F_0))) \tag{4}$$

where $F_m$ is the feature transformation of $F_0$ after $m$ CARBs and $R_m(.)$ the residual representation of the m-th CARB. Then for a CARN with $n$ CARBs, the residual information

is defined as Eq. (5):

$$F_N = F_0 + F_n. \tag{5}$$

This nested residual structure has a major positive effect on the tasks. The whole model (CARN) utilizes contextual information to improve objective metrics within a small network depth. Simultaneously, this allows the reduction of the number of CARBs in CARN as much as possible. $N_{CARB} = 8$ in the benchmark model is set, to prove the efficiency and effectiveness of the setting in Section 4.

## Loss function

Multiple CNN-based SR models chose MSE as the optimization loss of their networks because MSE usually leads to higher scores in the evaluation metrics such as PSNR. The MSE Loss is defined in Eq. (6):

$$L_{MSE} = \frac{\sum_{i=1}^{rH} \sum_{j=1}^{rW} \left\{ I_{(i,j)}^{HR} - \theta_G(I^{LR})_{(i,j)} \right\}^2}{r^2 \times H \times W} \tag{6}$$

where $\theta_G(.)$ represents the network optimization process; r, H, and W denote the number of samples, the number of pixels of sample height, and width, respectively. It is mentioned in *Ledig et al. (2017)* that once the magnification exceeds 4, $I_{LR}$ outputted by an MSE-optimized network lacks high-frequency details and appears to be extremely smooth in terms of texture while maintaining high-objective metrics. This is not in line with the subjective thinking direction of human beings when judging whether the image is clear or not. In terms of objective metrics, PSNR or SSIM are not highly consistent with the perceptual quality and human subjective visual effects. *Lim et al. (2017)*; *Zhang et al. (2018b)*; *Zhang et al. (2018c)* used the L1 loss, as defined in Eq. (7), to improve the models' performance, as *Blau et al. (2018)* pointed out that the L1 loss could tolerate larger errors compared to the MSE loss. Moreover, in terms of texture, the L1 loss enables networks to reproduce more high-frequency details than the MSE loss.

$$L_{L1} = \frac{\sum_{i=1}^{rH} \sum_{j=1}^{rW} \left\| \theta_G(I^{LR})_{(i,j)} - I_{(i,j)}^{HR} \right\|_1}{r^2 \times H \times W}. \tag{7}$$

In most scenarios such as medical diagnosis, humans do not calculate the distance between two images pixel by pixel, but rather judge the structural similarity between them. A measure based on structural similarity was proposed in *Wang et al. (2004)*, as defined in Eq. (8):

$$L_{SSIM} = 1 - SSIM \left[ \theta_G(I^{LR}), I_{(i,j)}^{HR} \right] \tag{8}$$

where $SSIM(.,.)$ represents the structural similarity measure of two samples. A loss function $L_{CARN}$ for CARN is designed, as defined in Eq. (9), which is a linear combination of L1 loss, MSE loss, and SSIM loss.

$$L_{CARN} = \alpha L_{L1} + \beta L_{MSE} + \gamma L_{SSIM} \tag{9}$$

where $\alpha, \beta, and \gamma$ are the weights of L1 loss, MSE loss, and SSIM loss, respectively.

In the conducted experiments, it is clear that when $(\alpha, \beta, \gamma) = (0.8, 0.1, 0.1)$, CARN showed the best performance by grid search. Therefore, $L_{L1}$ is the main component of the total loss function, enabling CARN to show better convergence and ensuring better subjective visual effects of pathology image SR results. $L_{MSE}$ improves the pixel-level accuracy and reduces fluctuations in objective metric values. During the training process, the change of $L_{SSIM}$ is extremely small ($0.0974-0.0979$ in CARN's $2\times$ SR task), which not only enhances the structural similarity of the output results but also fixedly enlarges the difference between the testing samples and training samples to improve the model's optimization level.

## EXPERIMENT

### Dataset: bcSR

The dataset CAMELYON (*Litjens et al., 2018*) originated from two pathology challenges organized by the Diagnostic Image Analysis Group (DIAG) and the Department of Pathology of the Radboud University Medical Center in Nijmegen, the Netherlands in 2016 and 2017. The goal of these challenges was to evaluate new and existing algorithms for automatic detection and classification of breast cancer metastases in WSI images of breast cancer sentinel lymph node sections.

WSI is a technique for multi-scale digitization of traditional slides at various resolutions and is commonly used in the field of pathology cell images. High-performance WSI scanners are usually very expensive. It is a highly feasible, efficient, and relatively inexpensive strategy to use SR technology to enlarge LR images from WSI, replacing the process of WSI scanning HR images. Thus, data were collected from CAMELYON17 and a benchmark dataset bcSR was established to facilitate SISR applications on pathology images of breast cancer.

Deep CNN networks cannot directly use the WSI file format as experimental samples and require efficient sampling of information-rich and representative patches. *Li & Ping (2018)* sampled 400 initial WSI files within the CAMELYON dataset and provided 220,000 pre-sampled coordinate pairs and five levels of sampling resolution for cancer and non-cancer regions, respectively. 1,000 pre-sampled coordinates were selected from each of the two regions and they were sampled from the original level (level 0) of the WSI file with a block size of $1,024 \times 1,024$.

Considering that the cell tissue cannot cover the entire slice plane tightly when using certain coordinate positions for sampling to make a digital image, multiple areas without cell tissue will be acquired, such as sampling at the edge of the entire slice. Image areas without cellular organization have little effect in the SR task, and these areas appear white in the digital images. Consequently, to improve the content of useful information in the dataset, the procedure is as follows: To begin, convert all digital images to grayscale images; Afterwards, binarize the image with the grayscale threshold of 224; Finally, calculate the proportion of white pixels (with a grayscale range of 225–255) in each image, and delete images with more than 60% of the white pixel area.

On the basis of the previous step, the RGB mean value $R_i$ of all pixels in each image and the RGB mean value $R_{all}$ of all pixels in all images were separately calculated. All the

images were ranked according to the distance from $R_i$ to $R_{all}$. Finally, the 1,200 images with the largest distance were chosen as the experimental dataset, called bcSR. This operation further amplified the pixel differences between samples.

Figure 3 describes the construction process of bcSR. The newly constructed dataset is beneficial for improving the objective metrics of deep CNN networks, making networks observe more regions, and training diverse samples to enhance the SR capability of the networks. Meanwhile, compared with random sampling, the proposed sampling strategy significantly improves the efficiency of SR tasks.

## Training details

The bcSR was split using 1000 images as the training set and 200 images as the test set for all models. The mean values of R, G, and B channels in training set are [0.7204, 0.4298, 0.6397]. LR images were generated by bicubic down-sampling. SRCNN (*Dong, Loy & Tang, 2016*), SRGAN (*Ledig et al., 2017*), EDSR (*Lim et al., 2017*), RDN (*Zhang et al., 2018c*), RCAN (*Zhang et al., 2018b*), and SWD-Net (*Chen et al., 2020*) were selected as comparison models. The network depth of EDSR, RDN, and RCAN were appropriately adjusted to match the total amount of CARN's parameters (see Table 1).

For CARN, $N_{CARB} = 8$ was set; the filter size of all convolutional layers was $3 \times 3$, and the number of channels was equal to 64, except that the number of channels in the middle layer of the CA Retention was 8 (*Reduction* $= 8$). During training, all samples were randomly rotated by 90 degrees to achieve the purpose of data augmentation and split the training samples into $64 \times 64$ patches as input and $48 \times 48$ in the $3\times$ up-sampling task. The ADAM was used to optimize the network. The learning rate was initialized to $10^{-4}$ and halved after every $2 \times 10^5$ minibatch update.

We use PSNR and SSIM to evaluate the performance of all SR models. PSNR is usually used to measure the distance between the final processed result and the original image, and its unit is 'dB'. The larger the PSNR, the lower the distortion of result. SSIM is usually used to estimate the similarity between two digital images. The range of SSIM is [0, 1]. The closer the SSIM is to 1, the more similar the two images are. For the experiments of all models in quantitative evaluation, the random seeds were respectively set to 1, 7, 11, 18 and 1011 to conduct the 5 groups of experiments. The average value of all experiments was taken as the final result of objective metrics.

The proposed network was implemented on the torch 1.8.0 platform and used a NVIDIA GeForce RTX 3090 (24GB) and a NVIDIA TITAN Xp (12GB) for all experiments. SRCNN and SRGAN are not able to run on the used device due to the problem of the original code platform, and SWD-Net only provides a $2\times$ up-sampling model in the original code. Thus, this study re-wrote and re-implemented these models.

## Quantitative experiments

This study presents objective metric results on $2\times, 3\times, 4\times$, and $8\times$ up-sampling tasks for all models in Table 2. Compared with other methods, the CARN achieved the state-of-the-art PSNR/SSIM in all tasks and had an objective metric gain of 0.188/0.0203 over the second-place SWD-Net. It is worth noting that using bcSR to train deep CNN models such as RDN,

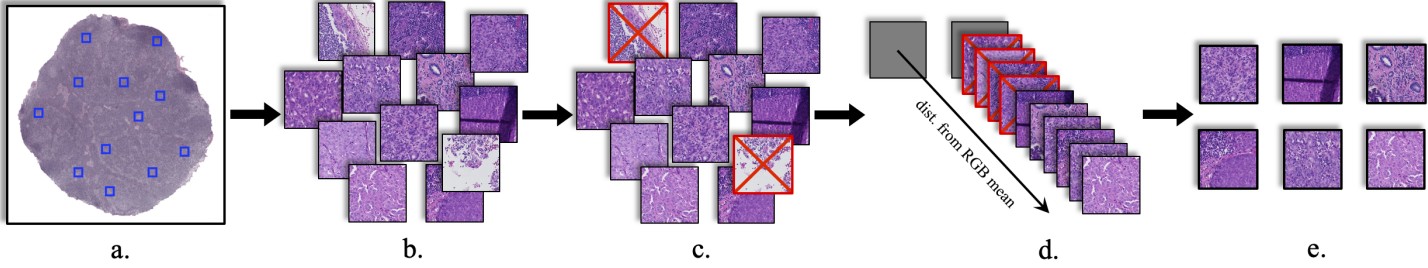

**Figure 3  Production process of bcSR.** (A) Select level 0 data in the CAMELYON dataset. (B) Randomly select coordinates and segment images. (C) Remove images with too many blank pixels. (D) Count the RGB channel means of all images and each image, and remove the images with the smallest distance. (E) Final results of bcSR.

**Table 1  The parameters of each model.**

| Model | bicubic | SRCNN | SRGAN | EDSR | RDN | RCAN | SWD-Net | CARN (ours) |
|---|---|---|---|---|---|---|---|---|
| Param. | – | 0.008 M | 6.097 M | 1.367 M | 2.375 M | 1.965 M | 3.158 M | 1.389 M |

RCAN, SWD-Net, and CARN, the objective metric results in the 2× up-sampling task surpass all current published SR works based on real-world images or pathology images, strongly demonstrating the adaptability of the bcSR in the SR tasks. RCAN and RDN are verified on Set5, Set14, B100, Urban100 and Manga109 datasets, and their best results come from Manga109. In 2× task, RCAN achieves 39.61/0.9788 and RDN achieves 39.38/0.9874 in PSNR/SSIM. The best result of EDSR comes from Set5: 38.20/0.9606. The performance of these competitive SR models in 3×, 4× tasks is lower than that of real-world image SR tasks: the best results of PSNR are higher than 34dB (3×) and 31dB (4×). Clearly, the SR task of histopathology images is more challenging than that of real-world images. In the 8× task, the performance of all models drops significantly. SWD-Net is specially designed for the SR task of histopathology images. The performance of the model on the 2× task in the HistoSR dataset (also from CAMELYON) is 32.769/0.9510, and the corresponding performance of EDSR is 32.676/0.9502. CARN can maintain a stable performance output in high-level upsampling tasks, and maintain an advantage in the reconstruction of texture details.

To demonstrate the effect of the loss function $L_{CARN}$, the suggested network was retrained with MSE loss (see CARN(MSE) in Table 2). The performance of the proposed model with MSE loss on 2×, 3×, or 4× SR task was slightly lower than the model itself. And the higher the magnification, the smaller the contribution of MSE loss is for SR image reconstruction. Furthermore, the CARN network was tested by adding a portion of samples—that have been removed during the bcSR construction, since they contain more than 60% blank area and have a short pixel channel distance—into the test set (see CARN* in Table 2). According to the results, the proposed network still shows similar or even better performance in these low-frequency regions, even though the removed images do not participate in the training process of CARN. This is exactly what people want to

**Table 2  Results of objective evaluation metrics.** The best results are marked in bold, the second results are underlined.

| Model | 2× | | 3× | | 4× | | 8× | |
|---|---|---|---|---|---|---|---|---|
| | PSNR | SSIM | PSNR | SSIM | PSNR | SSIM | PSNR | SSIM |
| bicubic | 35.247 | 0.9427 | 27.763 | 0.7655 | 27.019 | 0.6659 | 22.475 | 0.2776 |
| SRCNN | 35.522 | 0.9515 | 28.021 | 0.8117 | 27.475 | 0.7329 | 22.489 | 0.3624 |
| SRGAN | 35.998 | 0.9731 | 30.990 | 0.8244 | 28.606 | 0.7719 | 23.729 | 0.5580 |
| EDSR | 39.657 | 0.9728 | 31.095 | 0.8773 | 29.830 | 0.8058 | 24.366 | 0.5715 |
| RDN | 40.058 | 0.9762 | 31.240 | 0.8794 | 29.913 | 0.8074 | 24.392 | 0.5711 |
| RCAN | 40.144 | 0.9764 | 31.284 | 0.8812 | 29.916 | 0.8085 | 24.404 | 0.5749 |
| SWD-Net | 40.170 | 0.9639 | 31.186 | 0.8610 | 29.853 | 0.8000 | 24.465 | 0.5755 |
| CARN(MSE) | 40.336 | 0.9825 | 31.265 | 0.8986 | 29.888 | 0.8393 | 24.452 | 0.5739 |
| CARN | **40.358** | **0.9842** | **31.302** | **0.8995** | **29.964** | **0.8408** | **24.479** | **0.5763** |
| CARN* | 40.383 | 0.9837 | 31.297 | 0.8974 | 29.933 | 0.8407 | 24.484 | 0.5748 |

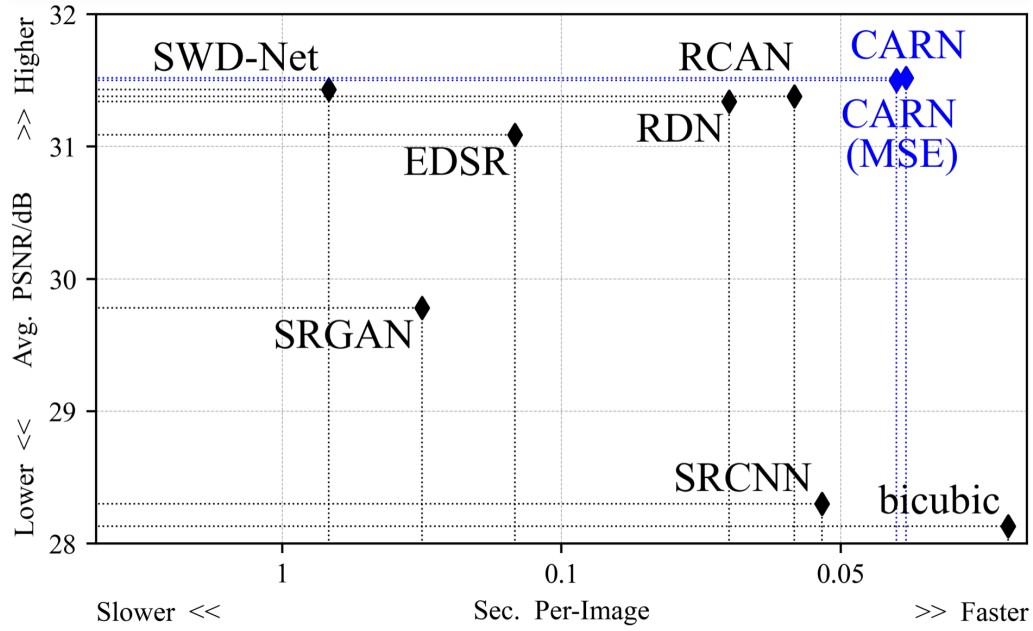

**Figure 4  The average time taken by all models to process an image.** In the 2× up-sampling task, the CARN spends 0.0283 s on each image. In the 4× up-sampling task, it takes 0.0342 s.

see. These results further demonstrate the satisfactory quality of this newly created dataset bcSR.

Figure 4 shows the average reconstruction time taken by all methods to process an image. The difference in efficiency mainly originates from the network depth and the optimization method. The suggested method is 29% faster than the second place in the 2× task, and the average time consumption is 15% faster than the second place in all up-sampling tasks.

## Qualitative experiments

Figure 5 shows the qualitative results of the CARN on 2× and 4× up-sampling tasks, respectively. The image details obtained by the proposed model far exceed other models, especially in some high frequency regions.

## Ablation experiments

### Influence of the number of CARBs

In Table 3, the results of bcSR on the 2× up-sampling task are shown with different numbers of CARBs. When the number of CARBs increases, the computational load of the network measured by FLOPs (*Molchanov et al., 2019*) augments significantly, and the network performance also improves. Among them, when the number of CARBs is 4, the network requires only 56.8% of the computation and 63.9% of the average time spent, achieving 99.4% of the PSNR results in the baseline model. On the other hand, when the number of CARBs is 12, the network requires 143.2% of the computation and 142.2% of the average time spent, reaching 100.2% of the PSNR result in the baseline model. Therefore, considering the performance of the comparative models, setting the number of CARBs to 8 is the best trade-off between the objective metric results and the model's training efficiency. Furthermore, a smaller number of CARBs enables CARN to exhibit optimal performance, approximately, which demonstrates the effectiveness of the model's substructure CARBs. In real medical diagnosis scenarios, a lightweight CARN with only several CARBs might be a convenient choice for maintaining the model's performance and real-time diagnosis efficiency.

### Influence of loss function

Figure 6 shows the effect of the loss function on the network's performance (PSNR) for all up-sampling tasks. It is clear that the convergence speed and final performance of CARN using the suggested combined loss are superior to those of CARN with commonly used MSE loss in all tasks. Since the area and spread of high-frequency information in pathology images are larger than those in general images, for SR tasks, the aforementioned loss performs better in terms of the model's convergence speed and objective metric results.

For CARN, exclusively using MSE loss will cause severe oscillation in the early training period, which is greatly obvious in the 2×, 3×, and 4× up-sampling tasks. Using the above-mentioned loss function enables CARN to have high stability throughout the training phase and maintain the same or even a faster convergence rate than the MSE loss.

Moreover, this study trained the CARN with the SSIM loss, only. The results show that the SSIM loss fails to converge the CARN: the PSNR on 2×, 3×, 4×, and 8× tasks are 12.341 dB (at epoch 119), 12.099 dB (at epoch 121), 12.124 dB (at epoch 19), and 4.072 dB (at epoch 1), which is quite different from the baseline results.

### Influence of reduction value in channel attention retention

The number of channels $C$ in the convolutional layers of the proposed baseline model is 64. Table 4 shows the effect of different *Reduction* values on the training process in the 2× up-sampling task. When *Reduction* = 8, namely $C$ becomes $\sqrt{C}$ through the following down-sampling convolutional layer in CARB, CARN obtains higher objective metrics. In

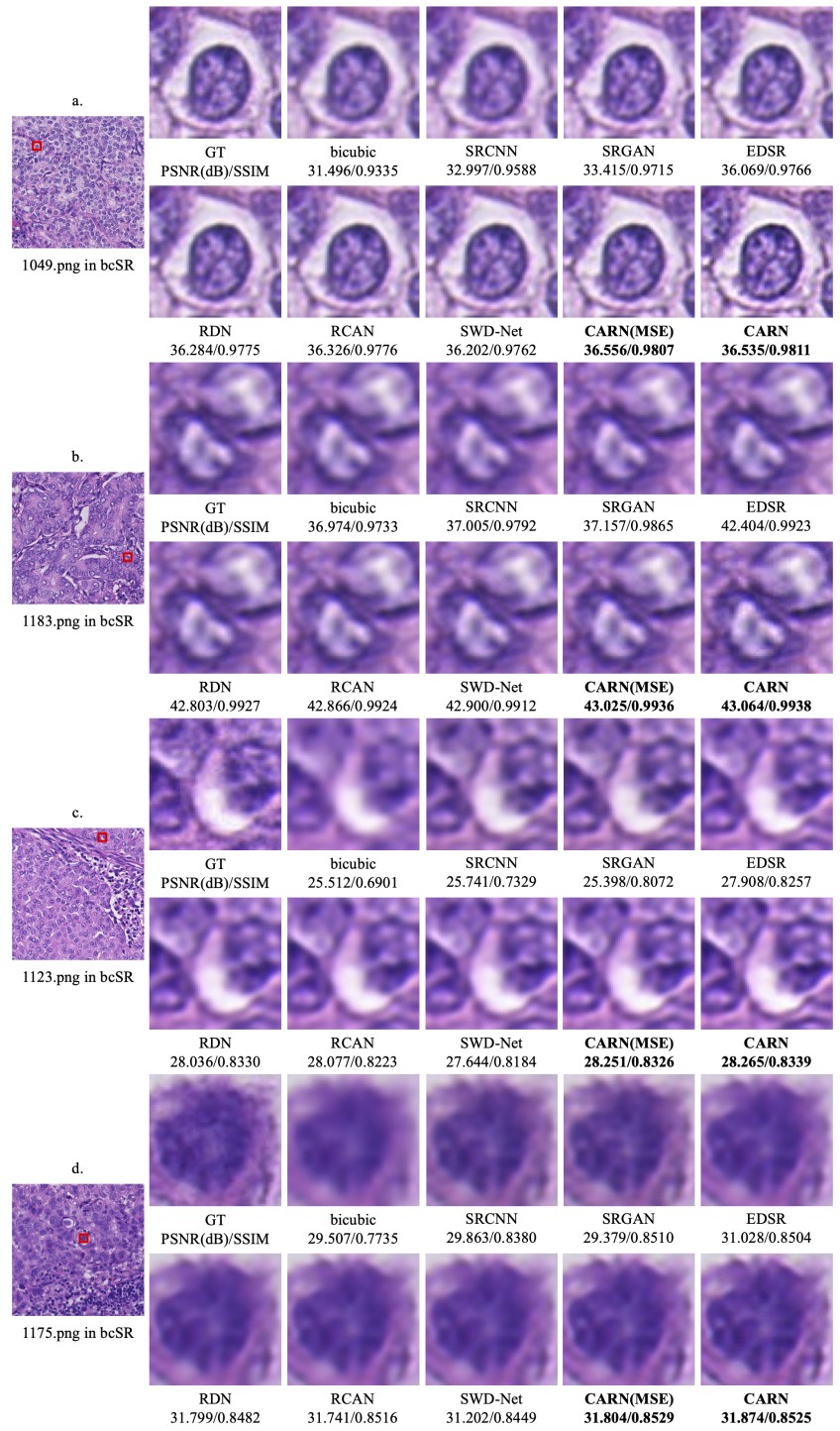

**Figure 5   Qualitative results.** (A) 1049.png in the 2× up-sampling task. (B) 1183.png in the 2× up-sampling task. (C) 1123.png in the 4× up-sampling task. (D) 1175.png in the 4× up-sampling task.

**Table 3 Investigations on the number of CARBs.**

| $N_{CARB}$ | 2 | 4 | 8 | 12 | 16 | 20 |
|---|---|---|---|---|---|---|
| PSNR(dB)/SSIM | 39.780/0.9526 | 40.137/0.9792 | 40.385/0.9839 | 40.446/0.9846 | 40.456/0.9845 | 40.513/0.9846 |
| FLOPs | 3.981 G | 6.421 G | 11.300 G | 16.179 G | 21.058 G | 25.936 G |
| Avg. Time (s) | 18.7 | 26.5 | 41.5 | 59.0 | 76.2 | 90.6 |

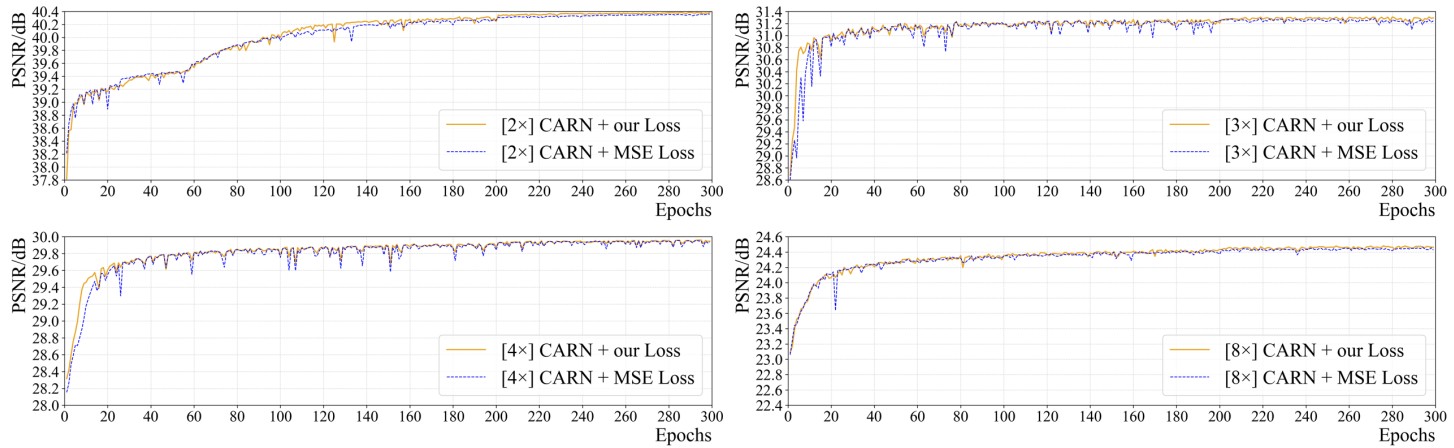

**Figure 6 The effect of two different loss functions on PSNR in the training process of CARN.**

**Table 4 The PNSR results of different *Reduction* values in the 2× upsampling task.**

| *Reduction* ($C = 64$) | 2 | 4 | **8** | 16 | 32 |
|---|---|---|---|---|---|
| PSNR(dB)/SSIM | 40.270/0.9792 | 40.262/0.9817 | **40.367/0.9832** | 40.291/0.9835 | 40.247/0.9830 |
| *Reduction* ($C = 128$) | 4 | **12** | 16 | 32 | 64 |
| PSNR(dB)/SSIM | 40.353/0.9777 | **40.421/0.9813** | 40.406/0.9797 | 40.271/0.9805 | 40.089/0.9794 |

fact, when the *Reduction* value is extremely small (such as 2 or 4), the network does not fully converge under the same training time constraint. In other words, CARN is not able to achieve a comprehensive performance gain on bcSR in terms of accuracy and efficiency when the *Reduction* value is small. Moreover, this study found that when $C$ is set to 128 and *Reduction* = 12, higher objective metrics are acquired. At this time, the value of *Reduction* is also close to $\sqrt{C}$.

## Results on binary classification for image diagnosis task

To further evaluate the role of CARN in the diagnosis task, this study chose ResNet-50 (*He et al., 2016*) as the baseline classifier and conducted the histopathology image classification task on the 2× SR results of the bcSR and PCam (*Veeling et al., 2018*) datasets. ResNet is one of the classic structures in the field of computer vision and has been widely used for tasks such as object classification. The PCam dataset is derived from Camelyon16, which contains 327,680 patches extracted from slides of 10× magnification (at 0.972 μm per pixel). Their size is 96 × 96. In the classification task, considering the input size of

**Table 5  Results on the binary classification task of breast cancer pathology images using ResNet-50.**

|  | Acc. | |
|---|---|---|
| **SR Model** | **bcSR** | **Pcam (*Veeling et al., 2018*)** |
| GT | 91.83% | 90.03% |
| bicubic (2× ↓) | 62.89% | 63.22% |
| bicubic | 79.53% | 66.40% |
| SRCNN | 77.76% | 68.87% |
| SRGAN | 81.04% | 70.35% |
| EDSR | 85.61% | 77.83% |
| RDN | 87.49% | 80.24% |
| RCAN | 87.31% | 83.10% |
| SWD-Net | 89.47% | 84.42% |
| CARN | **91.09%** | **87.20%** |

the ResNet-50 network, each image was segmented in bcSR into 64 images of $128 \times 128$ size. These segmented images maintained the same class labels as the original ones. The classification results are shown in Table 5. The deep learning-based SR models both achieved higher gains in performance than bicubic. Notably, CARN achieved the best results in both the bcSR and PCam datasets, with accuracies 11.56% and 20.80% higher than bicubic, and 1.62% and 2.78% higher than the second-ranked SWD-Net. The results for the image classification task show that the aforementioned CARN is able to recover more details from pathology images, thus facilitating downstream tasks such as real-time image diagnosis.

## CONCLUSION

In this study, a high-quality benchmark dataset bcSR was extracted from the public dataset CAMELYON for the pathology image SR tasks and developed an SR deep learning model (CARN). For dataset construction, WSI image files were collected in raw format from the public dataset, the format was converted and the images that are easy to use for SR deep models were filtered out. In the process of making the dataset bcSR, we expect that the set of selected images has the largest pixel distance, which can force the SR model to focus on more regions and benefit the training process of the SR model. Considering that most of the classical SR deep learning models are designed for real-world images, the characteristics of the real-world images and pathology images were analyzed, and a more effective CARB as well as a linear combination loss function were designed in the proposed model, CARN. Based on the above design, CARN can better transfer the information obtained in other types of images to histopathological images. The model was verified by multiple comparative experiments. Experimental results indicate that CARN achieves state-of-the-art performance on multiple upsampling tasks, with a boost of 0.18/0.0203 in the 2× SR task. Ablation experiments demonstrates that CARN makes a satisfactory trade-off between accuracy and efficiency. CARN improves the accuracy of image classification networks for diagnosis by 1.62%/2.78% on two datasets of different scales. In summary, bcSR and CARN provide effective priors for SR tasks of multiple

pathology images such as breast cancer and supply an efficient and robust solution for real medical diagnosis scenarios.

### Funding

The authors received no funding for this work.

### Competing Interests

The authors declare there are no competing interests.

### Author Contributions

- Feiyang Jia conceived and designed the experiments, performed the experiments, analyzed the data, performed the computation work, prepared figures and/or tables, and approved the final draft.
- Li Tan analyzed the data, prepared figures and/or tables, authored or reviewed drafts of the article, and approved the final draft.
- Ge Wang analyzed the data, prepared figures and/or tables, authored or reviewed drafts of the article, and approved the final draft.
- Caiyan Jia conceived and designed the experiments, analyzed the data, authored or reviewed drafts of the article, and approved the final draft.
- Zhineng Chen conceived and designed the experiments, analyzed the data, authored or reviewed drafts of the article, and approved the final draft.

### Data Availability

The bcSR dataset is available at Figshare: Jia, Feiyang (2022): bcSR. figshare. Figure. https://doi.org/10.6084/m9.figshare.21155584.v1

CARN: The Super-Resolution model proposed in the article is available at Github: https://github.com/MoyangSensei/CARN-Pytorch.

The bicubic algorithm is available in the Supplementary File.

The third-party computer codes are available at Github:

- The SRCNN is available at GitHub: https://github.com/yjn870/SRCNN-pytorch; "Accelerating the super-resolution convolutional neural network".

- The SRGAN is available at GitHub: https://github.com/tensorlayer/srgan; "Photo-realistic single image super-resolution using a generative adversarial network".

- The EDSR/RDN/RCAN are available at GitHub: https://github.com/sanghyun-son/EDSR-PyTorch (this code provides the implementation of RDN and RCAN); (EDSR): "Enhanced deep residual networks for single image super-resolution"; (RND): "Residual dense network for image super-resolution"; (RCAN): "Image super-resolution using very deep residual channel attention networks".

- The SWD-Net is available at GitHub: https://github.com/franciszchen/SWD-Net; "Joint spatial-wavelet dual-stream network for super-resolution".

The PCam dataset used for Binary Classification for Image Diagnosis Task is available at Github: https://github.com/basveeling/pcam.

## Supplemental Information

Supplemental information for this article can be found online at http://dx.doi.org/10.7717/peerj-cs.1196#supplemental-information.

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
