# Peer review of "A super-resolution network using channel attention retention for pathology images"

_PeerJ Computer Science, doi:10.7717/peerj-cs.1196_

## Round 0.1 · original submission · Major Revisions

The reviewers find merits in the paper. You are required to revise the manuscript and address all comments and suggestions of the reviewers. Good luck

Reviewer 1 ·

Basic reporting

1 “two multiple increments” seems weird and confusing in L182. The authors should consider rephrasing it.
2 The authors should explain ‘db’ when they first appear in the manuscript.
3 I suggest that the CONCLUSION section should be improved to better reflect the quality of the work.

Experimental design

1 The (1) formula seems not correct, the authors should double-check it.
2 The authors should explain why they adopted bicubic algorithm in L168-L169. Did they compare it with other alternatives?
3 For ‘Loss Function’ section, I am wondering why the authors consider the combination of L1 loss, MSE loss, and SSIM loss. Did they consider other loss functions such as L2 loss? Also, it seems the authors find the best weights for these three loss functions(0.8, 0.1, 0.1). The authors should provide information regarding weight space for these loss functions and corresponding performance metrics for each combination.
4 For “Dataset: bcSR” section, the authors should explain why they did preprocess steps in L287-L291. Is there any reference or experiments analysis for supporting it?
5 The authors did data augmentation in their experiments. Now that the datasets they generated are small, why did they pick only 1200 images as the experimental dataset?
6 The authors re-wrote and re-implemented some published models to compare, and even adjusted some models to match the total amount of CARN’s parameters. The authors should provide relevant details and codes for these models. Also, I am doubting the necessity of adjusting models to match the total amount of CARN’s parameters. Will adjusting models worsens those published models terribly?
7 For the hyperparameter optimization part, the authors didn’t consider a comprehensive hyperparameter combination space to experiment. For example, the authors didn’t explore the effects of the filter size of the convolutional layer and the number of channels on the predictive metrics in L307-L312.
8 Also, the data augmentation is not adequate. There are much more data augmentation strategies.

Validity of the findings

1 The authors should provide the datasets they created in the manuscript.
2 The authors should provide a summary regarding published SR models they tested and their corresponding performance for L324-L326. Also, I am curious about the result of 3×, 4×, 8× in L324-L326. The authors should provide them.
3 The authors are suggested to provide the results of all experiments they did as supplemental tables.
4 There is a lack of statistical analysis throughout the manuscript.
5 Other strategies such as cross-validation can provide a more unbiased evaluation rather than the three groups of experiments the authors picked in L315.

Reviewer 2 ·

Basic reporting

Exposition:
The paper is written in clear English and is understandable. Structure and references are sufficient.

Code:
The code is simple to understand but is missing comments in a few places. Could be more functional and less imperative. It is also unnecessarily reimplementing common utilities like image augmentation, conversions (np array to tensor), data loading, common transformations etc. which bloats the repo. As a side note: likely half of the code could be factored out into tested python utils or abstracted.

Experimental design

The metric comparisons and tests seem to be sufficient to prove that the proposed architecture in fact improves on key metrics.

I see no issues with the experimental setup.

Validity of the findings

Although the architectural innovations are very minor and around 50 lines of python code (excluding utils), the results and metric improvements are significant (often 2-4%).

It is likely that these gains will soon be eaten away by a more general innovation in image models (CNNs or transformers) that will then be fine-tuned to pathology, but that's just as an aside. As the paper stands right now, the findings are valid.

Additional comments

The code could be simplified and many functions could be replaced by OSS utils instead of reimplementing them.

---

## Round 0.2 · accepted · Accept

One of the reviewers acknowledged that the manuscript has been modified and comments have been incorporated successfully. As the Academic Editor, I have also checked the corrections made by the authors. I am satisfied with the revision and find it suitable for publication in its current form. Thank you for your fine contributions.

Reviewer 1 ·

Basic reporting

My concerns are addressed by the authors.

Experimental design

My concerns are addressed by the authors.

Validity of the findings

My concerns are addressed by the authors.

Additional comments

My concerns are addressed by the authors. The manuscript has been improved a lot.